# Model-Based Evaluation of Air-Side Fouling in Closed-Circuit Cooling Towers

**Björn Nienborg** [1,*] **, Marc Mathieu** [2]**, Alexander Schwärzler** [2]**, Katharina Conzelmann** [2] **and Lena Schnabel** [1]

1   Fraunhofer ISE—Institute for Solar Energy Systems, Division Thermal Systems and Buildings, Heidenhofstr. 2, 79110 Freiburg, Germany; lena.schnabel@ise.fraunhofer.de

2   Dr. O. Hartmann GmbH & Co. KG, Uhlandstrasse 30, 71665 Vaihingen an der Enz, Germany; MMathieu@dr-hartmann-chemie.eu (M.M.); ASchwaerzler@dr-hartmann-chemie.eu (A.S.); KConzelmann@dr-hartmann-chemie.eu (K.C.)

*   Correspondence: bjoern.nienborg@ise.fraunhofer.de; Tel.: +49-761-4588-5883

**Abstract:** Fouling is a permanent problem in process technology and is estimated to cost 0.25% of the gross national product. Evaporative cooling systems are especially susceptible to air-side fouling: as they work with untreated outside air, they are exposed to both natural (e.g., pollen) and human-made (e.g., industrial dust) contaminants. In addition, suspended solid particles and dissolved salts in the spray water are an issue. In this study we analyzed an approach for fouling detection based on a semi-physical (grey-box) cooling tower model which we calibrated with measurement data. A test series with reliable laboratory data indicates good applicability of the model. In three datasets, the performance decreases due to fouling (scaling, which was provoked intentionally) in the range of 5–11% were clearly detected. When applied to measurement data of two cooling towers in real applications, the model also proved to be well calibratable with relatively little data (two to four operating days). For two data sets, the model yielded reasonable results when applied to long term data: a cooling tower cleaning could be retraced and nominal operation was verified during the remaining time. During the analysis of a third data set a temporary performance deviation was found, which could not be explained with the recorded data. Thus, the approach turned out to be generally applicable but requires further verification and refinement in order to increase the robustness. If successful, it can be transferred to a commercial product for need-oriented maintenance in order to reduce cooling tower operating costs and environmental impact.

**Keywords:** fouling; scaling; closed cooling tower; performance; heat transfer rate; fouling resistance

## 1. Introduction

Fouling is a persistent issue in process engineering. Over 90% of heat exchangers in industry suffer from it [1]. The estimated costs due to oversizing, downtime and increased energy consumption sum up to 0.25% of the Gross Domestic Product in industrialized countries [2]. Evaporative cooling systems, which are frequently used in refrigeration systems and industry processes to dissipate condenser or waste heat to the environment, are also subject to fouling due to contaminants in the ambient air (e.g., dust, pollen) as well as suspended and diluted solids in the spray water.

Most publications on the effect of fouling on cooling tower performance refer to open cooling towers, their fills or externally connected heat exchangers. In this context, numerous publications are dedicated to the modelling and prediction of fouling processes and their effect on the thermal performance of cooling towers, e.g., Khan and Zubair study the effect of fouling on counter flow wet cooling towers [3]. They develop a fouling model, validate it with measurement data from previous publications literature and use it to evaluate the performance reduction due to fouling in different cooling towers geometries. Similarly, Cremaschi and Wu measured the impact of fouling on a condenser connected to an open cooling tower and use this data to evaluate their fouling model performance [4].

For large scale open cooling towers, methods for monitoring fouling are already available. A system presented by the energy provider Exelon compares measured data to predicted data based on characteristic curves derived from manufacturers' data [5]. The company Suez refers to the continuous monitoring of water chemistry and test tube sections operated under reference conditions in this context [6] (chapter 36). Successful fouling detection for an industrial size open cooling tower is also reported by Jin et al. [7]. They calibrate their model with operation data of the cooling tower under clean conditions and quantify the performance degradation due to fouling over time by comparing the modelled performance of a clean cooling tower with the measured real performance.

Information regarding the influence of air-side fouling on the performance of closed cooling towers available in literature is scarce, lacks information on the operating conditions as well as cooling tower geometries and shows large variations: while Qureshi reports a performance decrease of ~5% for a scaling thickness of 0.1 mm on the air-side of the tube bundle, which is the same order of magnitude specified by CTI, Hartvig claims it to be around 30% under these conditions [8–10]. At 1 mm scaling thickness, the literature reports performance reductions in the range of 20% to 35% (Hartvig does not specify results for this thickness) [8,9,11]. Zaza et al. model the scaling process on different types of cooling tower tubes in good agreement with their experiments on laboratory scale [12]. They determine performance decreases in the range of 3% to 4% during their tests but provide no information on the layer thickness. Despite the broad range of values reported it is very clear that air-side fouling can lead to a significant decrease in cooling tower performance with the negative consequences stated above. A graphical method for performance monitoring and detecting fouling based on its degradation which also seemed promising for smaller closed circuit cooling towers was found unsuitable in practice [13,14].

Therefore, the objectives of this paper is to fill the knowledge gap concerning closed cooling towers found in the available publications with the following steps: (a) provide reliable measurement data of a closed cooling tower which is subject to fouling/scaling on a laboratory scale, (b) employ a model-based approach with minimum need of sensors to detect the performance degradation due to fouling and (c) test the applicability of this approach to real installations based on field measurements.

## 2. Methodology

### 2.1. Cooling Tower Lab Tests

In order to generate detailed measurement data, a market available closed counter-flow cooling tower (Gohl VK8/5, specifications see Table 1) was continuously monitored during operation on a test rig. On the process fluid side, the volume flow could be varied up to 5300 L/h and the inlet temperature into the cooling tower up to 60 °C. On the air side, the cooling tower operated with ambient air, so temperature and humidity could not be regulated. However, the fan speed was regularly varied between 25%, 50%, 75% and 100% via a frequency converter. The spray pump could be switched on or off but not controlled. Figure 1 shows a scheme of the setup.

**Table 1.** Key characteristics of the investigated cooling tower.

| Description | Unit | Value |
|---|---|---|
| Nominal capacity | kWth | 35 |
| Fan electric power | kWel | 1.5 |
| Spray pump electric power | kWel | 0.7 |
| Nominal air volume flow rate | $m^3/h$ | 5350 |
| Vertical passes | - | 18 |
| Horizontal tubes | - | 10 |
| Tube distance hor./vert. | mm | 30 |
| Tube diameter | mm | 26.3 |

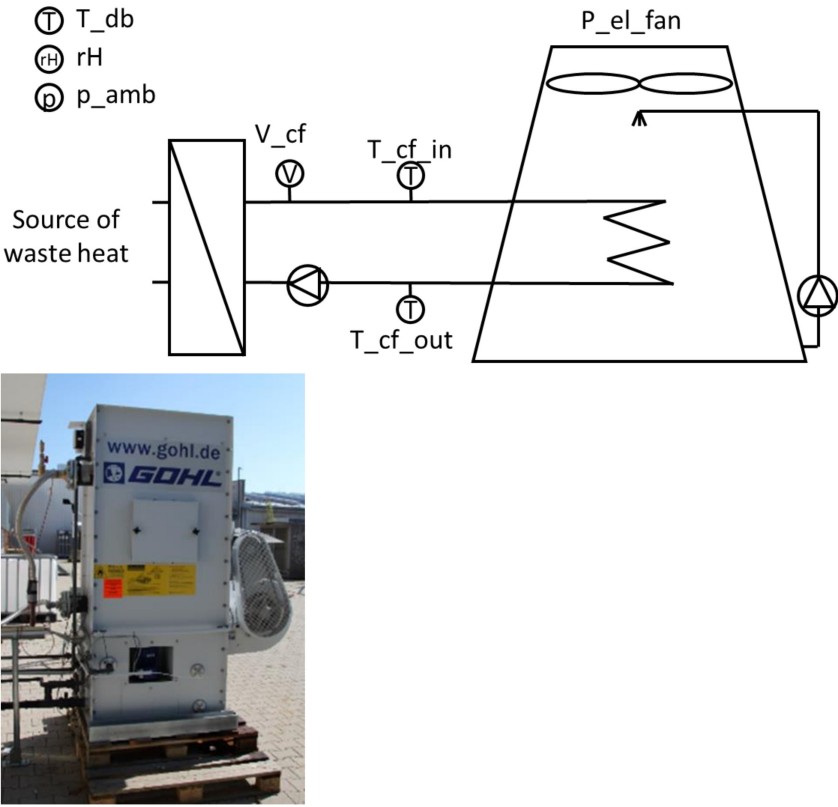

**Figure 1.** Simplified scheme of the cooling tower test setup including sensor positions; ambient air conditions were measured next to the cooling tower.

The installed sensors record values every minute. Three measuring points are located in the cooling circuit (inlet/outlet temperature and volume flow). The supply air conditions (temperature, humidity, pressure) as well as the operating status of the spray pump (on/off) and the electrical power consumption of the fan are measured. The most important technical data of the measurement technology are listed in Table 2. The combined temperature and humidity probe is covered by an actively ventilated radiation shield produced by Thies Clima.

**Table 2.** Description and specifications of the installed sensors.

| Description | Type of Sensor | Max. Error | Resolution |
|---|---|---|---|
| Dry bulb temperature | Combined T/H-Probe (Pt100 + | +/− (0.3 K + 0.3 K) * | 0.02 K |
| Relative humidity | capacitive sensor) | +/− (2% + 0.3%) * | 0.02% |
| Air pressure | Piezo-resistive pressure sensor | +/− (1 h Pa + 0.5 h Pa) * | 0.04 h Pa |
| Temperatures cooling fluid (in/out) | Pt100, 4-wire | +/− 0.05 K | 0.005 K |
| Volume flow cooling fluid | magnetic-inductive | +/− (0.5% + 0.3%) * | 0.1 L/h |
| Fan electric power | | +/− 1.5% | 10 mW |

* (sensor + data acquisition).

A measuring device "Blomat" by Dr. Hartmann Chemietechnik controls the blow-down based on the conductivity of the spray water. The device additionally monitors the pH value and the redox potential. Biofouling is controlled by regular addition of biocide. Naturally, the requirements of the latest German Federal Emissions Control Regulation, such as biweekly quick test for Colony Forming Units (CFU) and quarterly legionella sampling by an accredited laboratory, were also complied with.

### 2.2. Cooling Tower Field Measurements

In order to collect further operation data, two cooling towers in the range of 550 to 910 kW nominal capacity were monitored during operation in their respective applications

(dissipation of waste heat from chillers and free cooling). The same sensors as in the laboratory tests were selected for measuring the ambient conditions and fluid temperatures. The electricity consumption could not be measured in these cases, instead the fan signal (2 speeds) was recorded. The volume flow of the process fluid (water-glycol) was recorded with ultrasonic clamp-on meters. An overview on the specifications of the installations is given in the following:

- CT1: Closed circuit wet cooling tower with air inlet and outlet at the top, design capacity 910 kW, nominal fan power 5.3 kW/21.4 kW,
  - Water-glycol flow measurement with Flexim Fluxus F601, automatically switching between transit time and Doppler method due to insufficiently bled piping
  - Water-glycol temperature measurements with immersed 4-wire-Pt100,
  - T/H-probe as specified above but with passive radiation shield Davis PN 7714 (best results for passive shields in a comparative study by the World Meteorological Organization [15])
- CT2: Closed circuit wet cooling tower, design capacity 550 kW, nominal fan power 3.3 kW/14.0 kW,
  - Water-glycol flow measurement with Systec deltawaveC,
  - Water-glycol temperature measurements with contact sensors (3-wire-Pt100),
  - Ambient conditions measured as with CT1.

### 2.3. Data Treatment

As the measurements contain unsteady data due to changes in set values and ambient conditions, quasi-stationary data is extracted for all model calibrations. For this purpose, the standard deviation is determined for five consecutive measurements (three before the actual measurement and one after) for

- ambient temperature;
- water inlet temperature;
- flow rate;
- fan signal.

The utilized data is then restricted to all values with a standard deviation lower than twice the average over all standard deviations.

### 2.4. Cooling Tower Calculation Model

For the data evaluation a cooling tower calculation model based on the Merkel theory and effective NTU method was employed [16]. As it reduces the heat and mass transfer within the cooling tower to a 1D-problem it requires little computational resources. At the same time, it has shown decent accuracy in past studies [17,18]. The relevant assumption of this model is that the heat transfer process on the air and water side can be simplified in such way that they basically only depend on the respective mass flow and operating conditions, as shown in Figure 2 and Equations (1)–(3).

$$Qu_{aint} = 1/R_{int} = \delta \, ; \, \dot{m}_w^{0.8}/\mu_w^{0.5} = f(\dot{m}_w, \, t_{w,in}) \tag{1}$$

$$UA_{ext} = 1/R_{ext} = \gamma \, ; \, cp_{sat} \, ; \, \dot{m}_a^{0.8} = f(\dot{m}_a, \, cp_{sat}) \tag{2}$$

$$cp_{sat} = (h_{a,out} - h_{a,in})/(T_{wb,out} - T_{wb,in}) \tag{3}$$

From these heat transfer coefficients, the effective cooling capacity is then calculated following the ε-NTU method. All the required input values except the air mass flow were directly measured or calculated within the model. The air mass flow cannot be measured accurately without either imposing a significant pressure drop (and thus influencing the cooling tower performance) or costly measurement technology (e.g., laser Doppler anemometer). Therefore, it was derived from the control signal of the fan. As the exact fan curve is not usually known, an additional offset to the fan signal was introduced and

calibrated together with the respective coefficients for air and water side heat transfer properties ($\gamma$ and $\delta$, respectively), which are specific for every cooling tower.

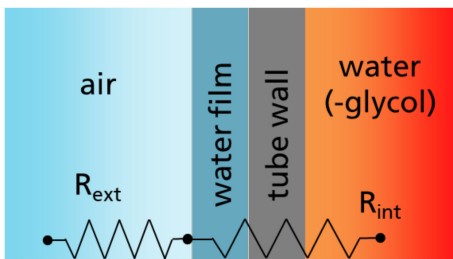

**Figure 2.** Schematic of the simplifications applied by the selected cooling tower model.

The model was implemented in Python 3.1. Humid air properties were determined by Cool Prop [19]. The calibration was performed by minimizing the sum of the squared difference between measured and calculated power of the selected data set. It was realized in Python with the L-BFGS-B algorithm available in the function *minimize* of the package scipy [20].

## 3. Results

In the following the workflow and results of the measurements are presented and discussed. The first subchapter concentrates on the laboratory tests. The second part focusses on the field measurements and thus the applicability of the method in practice. All presented performance measurements correspond to one minute values.

### 3.1. Results of Cooling Tower Lab Tests

3.1.1. Description of Operating Period

Measurements were carried out from September 2017 until September 2018 and from May 2018 until November 2019, with short downtimes due to minor revisions in the setup or annual closing. Initially the process fluid loop was operated with pure water as heat transfer medium. In the end of November 2017, it was replaced by a glycol mixture for frost protection. After a leakage in July 2018 water was used as heat transfer fluid again and operation continued until September. Since the scaling progress was much slower than expected with city water, the concentration factor was varied over time and the deposition of carbonate was accelerated by adding $NaHCO_3 + CaCl_2$. Table 3 contains an overview on all experiments.

**Table 3.** Overview on executed lab experiments.

| Period (DD.MM.YY) | Objective of Experiment | Makeup Water Composition | Dataset Code |
|---|---|---|---|
| 03.09.17–13.09.17 | Reference measurements | Decarbonized water | Lab-Cal1 |
| 14.09.17–27.11.17 | Scaling test 1 | City water | Lab-Scal1a |
| 30.11.17–09.07.18 [1] | Scaling test 1 | City water + $NaHCO_{3+} CaCl_2$ [2] | Lab-Scal1b |
| 13.07.18–10.09.18 | Scaling test 1 | City water + $NaHCO_{3+} CaCl_2$ [2] | Lab-Scal1c |
| 18.05.19 | Descaling | Descaling of tube bundle with acid, subsequent passivation | - |
| 05.06.19–28.08.19 | Scaling test 2 | City water + $NaHCO_{3++} CaCl_2$ [2] | Lab-Scal2 |

During the test period the thickness of the scaling layer was determined regularly with a Rotec BB20 magneto-inductive appliance. The results of four tubes at the top of the bundle (accessible via a service hatch at the side of the cooling tower housing, counting from the top) and the average over all ten tubes at the bottom (accessible via a second service hatch) for the first year of operation are plotted in the Figure 3. Figure 4 shows photos of the top tubes at different stages of scaling.

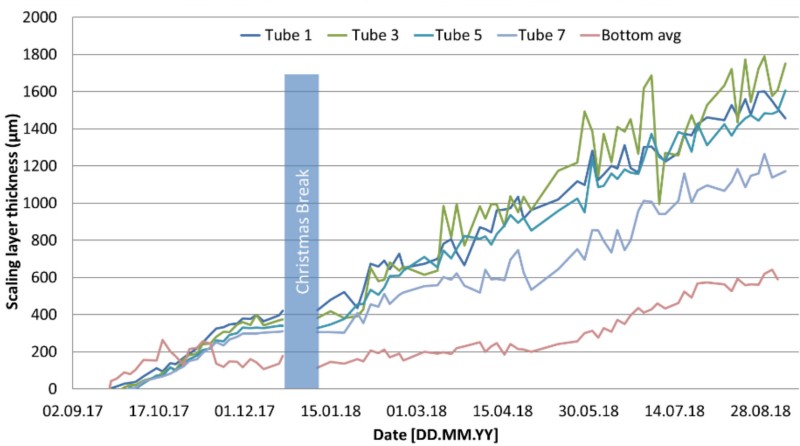

**Figure 3.** Thickness measurement at four tubes at the top and all bottom tubes (averaged) of the bundle.

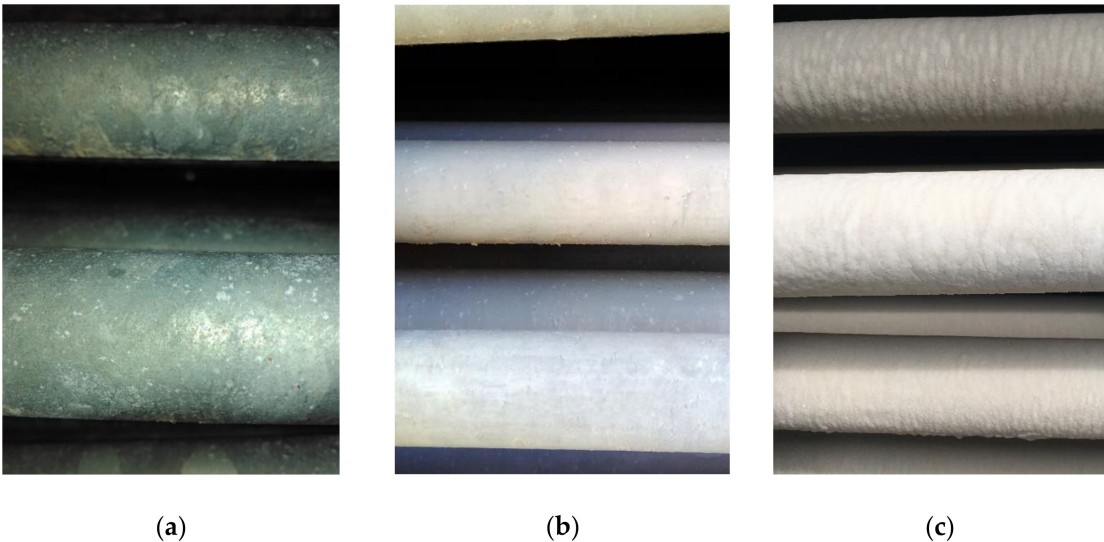

　　　　　**(a)**　　　　　　　　　　　　　　　**(b)**　　　　　　　　　　　　　　　**(c)**

**Figure 4.** Top tubes after measurements with softened water (**a**) and after 1 (**b**) and 12 (**c**) months of operation with hard water, respectively (Scaling test 1).

The cooling tower was emptied during the following winter (2018/19). Unintendedly, the inside of the tube bundle corroded during this period, leading to higher turbulence and thus pressure drop and heat transfer coefficient on the water side. After chemically removing the scale layer on the outside of the tubes, the scaling test was repeated in summer 2019 in order to validate findings.

[1] Operation with water glycol as heat transfer medium. [2] Addition of $NaHCO_3$ for accelerated scaling after January 2018, after April 2018 additionally $CaCl_2$.

As explained in Section 2.1, the laboratory cooling tower was operated at a test facility which allowed the control of the water flow rate and temperature as well as the fan speed. Ambient conditions were defined by the weather. An overview on the covered operation conditions during the respective tests is given in Table 4.

### 3.1.2. Results of First Scaling Test

The model was calibrated with the data recorded during the initial reference measurements (Lab-Cal1). Figure 5a shows the good agreement of the calculated (by the model) and measured (in the cooling water loop) powers. A total of 99.7% of the data points are within the +/−10% boundaries (black solid/dotted lines), 94.9% even have a deviation below +/−5%.

**Table 4.** Overview over the datasets of the laboratory cooling tower used for model calibration and evaluation; ranges are marked with "MIN ... MAX", steps are delimited with "/".

| Dataset Code | Ambient Temperature [°C] | Wet Bulb Temperature [°C] | Water Inlet Temperature [°C] | Water Flow Rate [m³/h] | Number of Data Points |
|---|---|---|---|---|---|
| Lab-Cal1 | 12.3 ... 27.5 | 10.5 ... 17.6 | 20.2 ... 36.1 | 3.3/4.2/5.2 | 5737 |
| Lab-Scal1a | −2.1 ... 29.5 | −3.3 ... 20.6 | 17.8 ... 39.8 | 3.9 ... 5.3 | 5161 |
| Lab-Scal1b | 12.5 ... 33.6 | 11.0 ... 21.3 | 26.3 ... 35.6 | 2.5 ... 4.2 | 30,205 |
| Lab-Cal2 | 13.7 ... 29.9 | 12.6 ... 20.5 | 26.2 ... 28.0 | 4.1 ... 4.2 | 1824 |
| Lab-Scal2 | 8.0 ... 39.9 | 7.5 ... 26.2 | 20.0 ... 32.1 | 3.9 ... 4.3 | 102,084 |

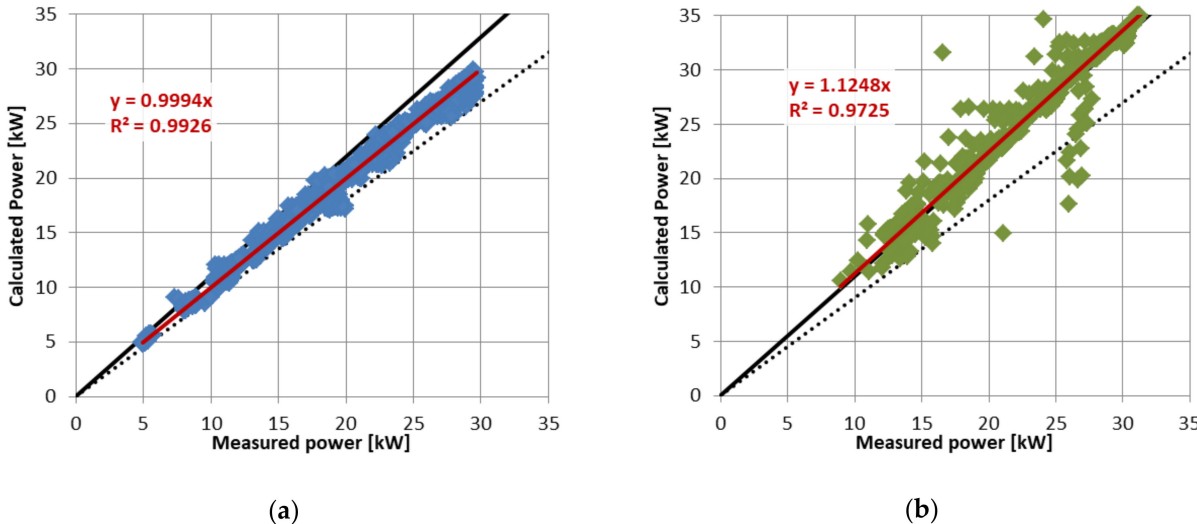

(**a**)   (**b**)

**Figure 5.** (**a**) Result of model calibration prior to scaling test 1 (dataset Lab-Cal1); (**b**) Comparison of measured and calculated cooling tower power after one year of operation (dataset Lab-Scal1b).

The calibrated model was then used to recalculate the values measured at the end of the first scaling test (dataset Lab-Scal1c). As Figure 5b indicates there is a clear deviation between the model and the measurements. From the inverse of the slope of the linear trend through the data points it can be derived that the degradation is about 11%. Considering a thickness of the scale layer of 1 mm (averaged over the entire bundle, see Figure 3) and assuming a linear behavior of the thickness on the performance reduction, this corresponds to a specific performance decrease of 11% per mm fouling layer. As the thermal resistance for a cylindrical layer depends on the natural log of the ratio of the external to the internal radius, the specified assumption is a simplification. The resulting values can only be compared for similar thicknesses or in terms of order of magnitude. A continuous evaluation of this period is not possible as the heat transfer medium was temporarily exchanged during the winter and spring 2018/19 (see Section 3.1.1 for explanation).

Thus, the period with a water-glycol mixture needs to be evaluated separately. For the graph in Figure 6 the model was calibrated with the initial measurements of the dataset Lab-Scal1b. Subsequently the relative deviation (difference between calculated and measured capacity divided by the measured capacity) was evaluated for entire measurement period. A continuous slight increase of the value, corresponding to a decrease in cooling tower capacity, can be observed. At the end of the regarded period the average decrease is about 5%. Considering a scale thickness of approximately 0.4mm and under the assumption of a linear correlation between the thickness and the performance change, this leads to a specific performance decrease of 12.5%/mm.

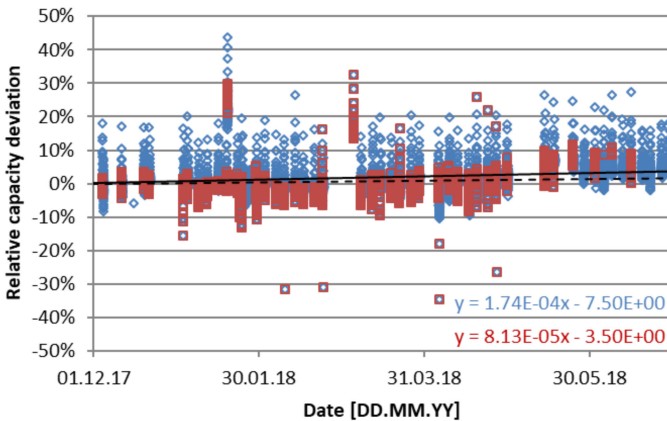

**Figure 6.** Trend of relative decrease of cooling tower power (calculated minus measures divided by measured) during the operation with water-glycol (data set Lab-Scal1a); model was calibrated with data recorded in the first week of December); blue dots: all fan speeds; red dots: only 100% fan speed; black lines: linear regression (solid: all fan speeds, dashed: only 100% fan speed).

### 3.1.3. Results of Second Scaling Test

Since the switching of the heat transfer medium did not allow a continuous evaluation of the scaling process in the first test, it was repeated. A significant difference to the first test was that the water flow rate was now limited to approximately 4 $m^3$/h since the tubes corroded on the inside during the operation pause. The positive effect was a significantly higher heat transfer coefficient confirmed by the model calibration. This time the measurement data of the first two operation days with city water were used for model calibration. An initial characterization with decarbonized spray water did not seem necessary as the slow scaling progress in the first test showed. Figure 7 shows the results of the calibration with 100% and 95.4% of the data points lying within the +/−10% and +/−5% boundaries, respectively.

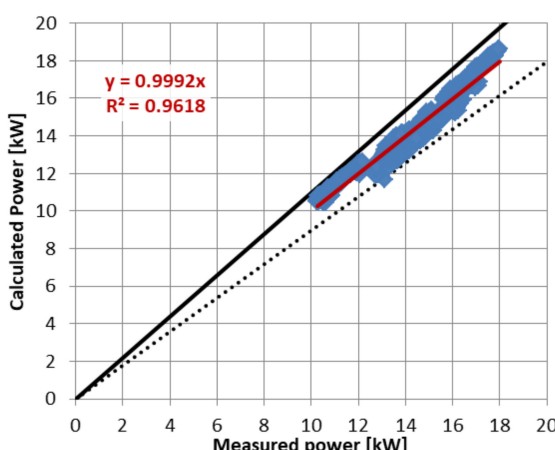

**Figure 7.** Result of model calibration prior to scaling test 2 (data set Lab-Cal2), measured vs. calculated thermal power of cooling tower; black solid/dotted lines show +/−10% boundaries.

Figure 8 contains information on the effect of the scaling on the cooling tower performance. Despite fluctuations the decrease in capacity is evident. At the end of the operation period the capacity dropped by approximately 8% compared to the model. As the average scale layer was only about 150 μm, the specific performance decrease was about 53% per mm fouling layer.

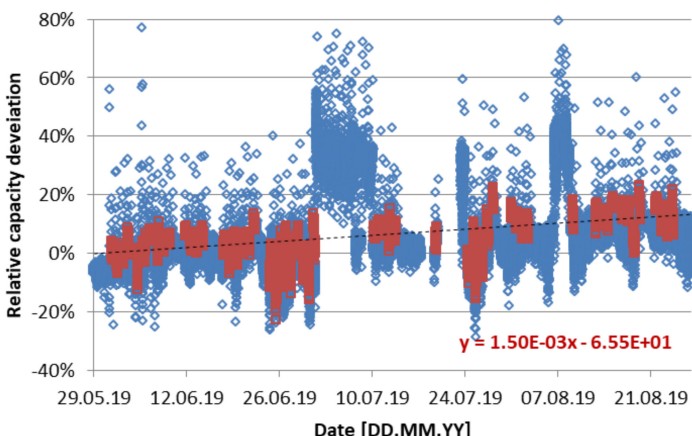

**Figure 8.** Trend of relative decrease of cooling tower power (calculated minus measured divided by measured) during scaling test 2 (data set Lab-Scal2); blue dots: all fan speeds; red dots: only 100% fan speed black dashed line: linear regression only 100% fan speed.

*3.2. Results of Field Measurements*

In the following the results are presented for the two monitored field cooling towers.

3.2.1. Results for CT1

For the interpretation of the results for CT1 the following points regarding its installation and operation must be kept in mind:

- The tower can either dissipate waste heat from a chiller or provide free cooling; the respective set temperatures are 20 °C and 8–11 °C. Therefore, the two step fan switches frequently in part load, leading to unsteady operation;
- For sound protection the air intake is located on the top of the tower, next to the outlet; thus, depending on the weather conditions, a varying recirculation can be expected which is not accounted for in the measurements;
- In fall and winter 2018/19 the cooling system of the building was refurbished; due to the downtimes and changes in operation the operation periods before and after may not be comparable;
- The tower is installed on the roof of a three-story building while the bleed valves are located next to the chiller in its basement; therefore the circuit is insufficiently bled which may partly lead to faulty flow measurements (the flow sensor switches automatically between transit time and Doppler method).

The following Table 5 gives an overview on the datasets used during the evaluation of the results. The cooling tower operated with a deficient spray water treatment and blowdown system until 11th of July which lead to visible scale depositions on the tube bundle and partly clogged spray nozzles. Therefore, it was mechanically cleaned by the manufacturer on July 12th/13th (no data for those days). The subsequent two days of measurement data were selected for the calibration of the model. As before, the calibration yielded good results with most data points inside the +/−10% boundaries (see Figure 9).

The application of the calibrated model to the measurement data during the respective summer clearly shows the effect of the cooling tower cleaning (Figure 10): the average deviation before July 11th is 3.9%, while there is no evident trend or deviation visible afterwards. Thus, it can be deducted that the effect of the cleaning is detectable. At full fan speed the scattering of the data mostly is within the +/−10% boundaries as during the calibration of the model. At the low fan speeds there are periods with significantly higher positive deviations. A possible explanation is that the cooling tower performance suffers from recirculation of air promoted by wind.

**Table 5.** Overview over the datasets of CT1 used for model calibration and evaluation; ranges are marked with "MIN . . . MAX".

| Dataset Code | Period [DD.MM.YY] | Ambient Temperature [°C] | Wet Bulb Temperature [°C] | Water Inlet Temperature [°C] | Water Flow Rate [m³/h] | Number of Data Points |
|---|---|---|---|---|---|---|
| CT1-Cal1 | 16./17.07.18 | 17.9 . . . 30 | 14.9 . . . 18.9 | 21.2 . . . 27.5 | 78.4 . . . 82.5 | 2872 (7.7%) * |
| CT1-Eval1 | 21.06.18–21.09.18 | 11.9 . . . 35.7 | 7.9 . . . 23.9 | 20.3 . . . 41.1 | 60.0 . . . 99.7 | 76,576 (9.6%) * |
| CT1-Cal2 | 26./27.05.19 | 14.8 . . . 24.6 | 11.7 . . . 15.8 | 21.4 . . . 26.5 | 27.9 . . . 46.8 | 1837 (67.9%) * |
| CT1-Eval2 | 19.05.19–02.02.20 | −1.9 . . . 39.0 | −2.4 . . . 25.5 | 7.0 . . . 40.3 | 18.8 . . . 56.2 | 176,789 (30.8%) * |

\* Share of data points at low fan speed.

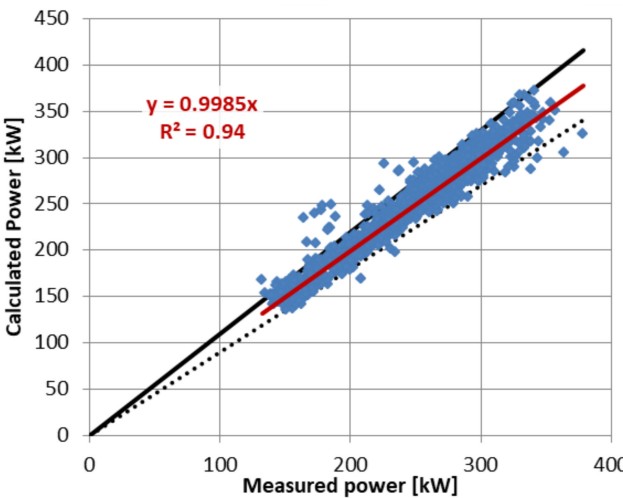

**Figure 9.** Result of model calibration with dataset CT1-Cal1, measured vs. calculated thermal power of cooling tower (2.2% outside +/−10% boundary); black solid/dotted lines show +/−10% boundaries.

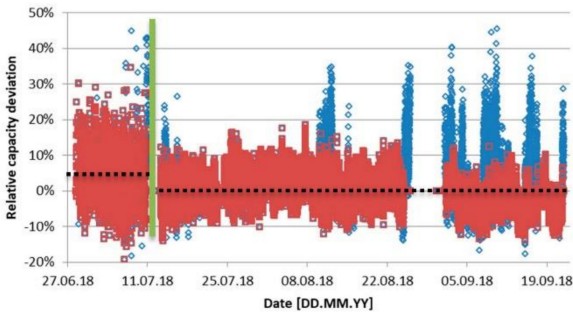

**Figure 10.** Trend of relative decrease of cooling tower power (calculated minus measured divided by measured) of dataset CT1-Eval1; blue dots: all data, red dots: only 100% fan speed, green line: cleaning, dotted black lines: average before/after cleaning.

Figure 11 shows the model applied to measurement data of the following year 2019. The lack of data during fall/winter is a consequence of the data treatment (see Section 2.3) excluding strongly unsteady operating conditions. The results show a general deviation of at least 20% compared to the 2018 data. This could be interpreted because of degradation. Yet, the predominant water flow rates during the shown period are significantly below the flow rates during calibration (see Table 5), so it is well possible that the model needs to be recalibrated. A result of a recalibration with measurement data from May 2019 is shown in Appendix A and proves that the offset can be avoided that way.

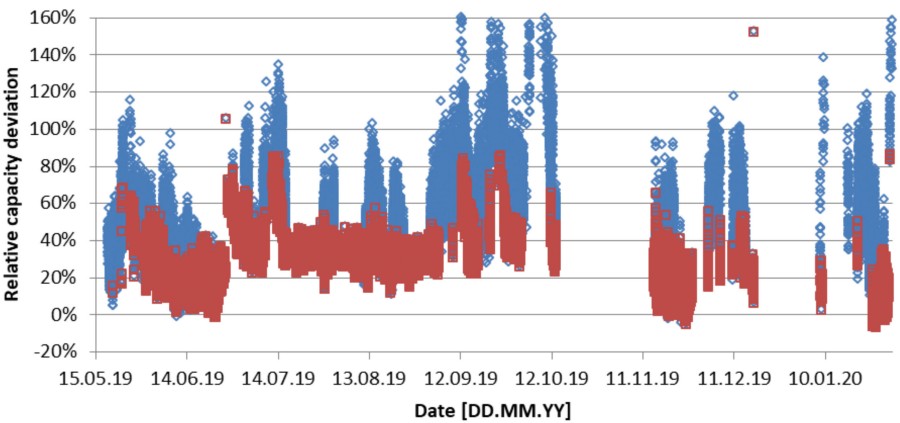

**Figure 11.** Trend of relative decrease of cooling tower power (calculated minus measured divided by measured) of dataset CT1-Eval2 calibrated with dataset CT2-Cal1; blue dots: all data, red dots: only 100% fan speed.

Another characteristic feature of the results with either calibration is that there seems to be a temporary offset (capacity deviation) of approximately 15% from July until October. Afterwards the results fall back to the initial order of magnitude. For this behavior no explanation could be found in the recorded data or in the operations diary of the cooling tower. This fact indicates that the model-based approach requires further investigation.

### 3.2.2. Results for CT 2

For the interpretation of the results for CT2 the following points regarding its installation and operation must be kept in mind:

- The tower dissipates waste heat from a chiller at a set temperature of 25 °C. Therefore, the two step fan switches frequently in part load, leading to unsteady operation;
- the tower is installed inside a building with an uninsulated air duct of approximately 8m length at the inlet; therefore, the air inlet conditions at the evaporative section of the cooling tower may differ somewhat from the ambient conditions measured outside the building;
- recirculation of air is expected to happen only under very specific wind conditions.

Table 6 summarizes the key data of the operating periods used for calibration and evaluation. The specified calibration period was selected in order to have a minimum share of operating points at low fan speed. The calibration worked with decent accuracy as Figure 12 shows. Only 2.4% of the data points are outside the +/−10% boundary. Yet, at the low fan speeds, two groups of points appear, which are off the ideal slope. Consequently, the coefficient of determination is low compared to the calibration results of the previous data sets.

**Table 6.** Overview over the datasets of CT2 used for model calibration and evaluation; ranges are marked with "MIN . . . MAX".

| Dataset | Period [DD.MM.YY] | Ambient Temperature [°C] | Wet Bulb Temperature [°C] | Water Inlet Temperature [°C] | Water Flow Rate [m³/h] | Number of Data Points |
|---|---|---|---|---|---|---|
| CT2-Cal1 | 11.07.19–15.07.19 | 14.5 . . . 28.5 | 12.3 . . . 20.2 | 26.1 . . . 33.2 | 81.6 . . . 85.2 | 4114 |
| CT2-Eval1 | 01.07.19–25.01.20 | −3.2 . . . 38.9 | −3.6 . . . 24.9 | 24.4 . . . 39.9 | 28.8 . . . 85.2 | 170,796 |

\* Share of data points at low fan speed.

Figure 13 presents the results for the model-based evaluation of CT2. For the period until the end of October, the model yields good results with a deviation at 100% fan speed (red dots) mostly oscillating around 0%. Starting from November there is not a significant number of evaluable data due to low ambient temperatures and the resulting transient cooling tower operation.

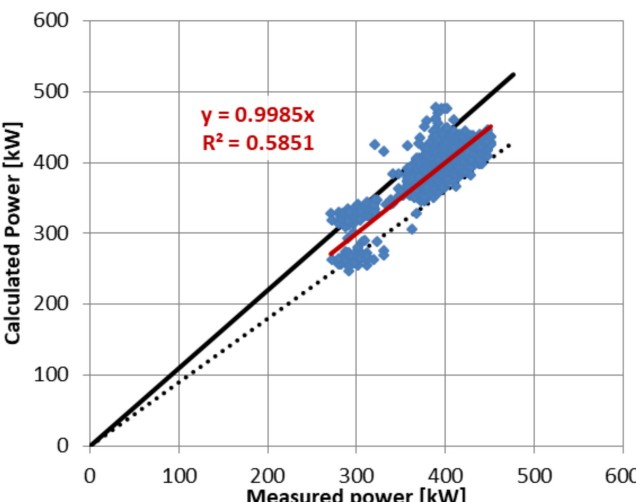

**Figure 12.** Result of model calibration with dataset CT2-Cal1, measured vs. calculated thermal power of cooling tower (2.4% outside +/−10% boundary); black solid/dotted lines show +/−10% boundaries.

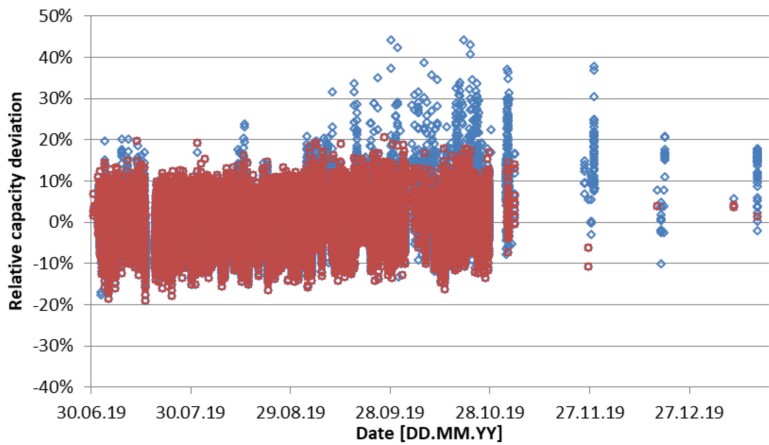

**Figure 13.** Trend of relative decrease of cooling tower power (calculated minus measures divided by measured) of dataset CT2-Eval1; blue dots: all data, red dots: only 100% fan speed.

## 4. Discussion and Conclusions

A model-based approach for detection of fouling and operation monitoring has been developed and tested under laboratory and field conditions.

To generate test data, a closed cooling tower at laboratory scale was operated and measured twice over a period of several months with intentional scaling. During the first campaign an average scale thickness of 1.1 mm (top/bottom tubes: 1.6 mm and 0.6 mm, respectively) was achieved. During the second run the test was terminated at 150 μm average thickness.

The cooling tower model could be well calibrated during both tests. The model-based evaluation of the performance degradation yielded around 15% drop for the first and 8% for the second campaign. Evidently the performance degradation due to fouling can be detected with the chosen approach. Results at full fan speed show the lowest scattering as the relative measurement errors and potential disturbances caused by the ambient (e.g., recirculation of air) is minimal.

Yet, the overall effect (and thus the achievable energy savings due to detection and cleaning) is strongly dependent on the specific operating conditions. In addition, there are periods with significant scattering in the results during continuous monitoring (especially in the second run). Possible causes are:

- (Temporary) defects not associated to the scaling of the cooling tower (e.g., partially blocked pump or nozzles);
- Thermal inertia due to sump water volume;
- Variations due to make-up water being added or blow-off being purged;
- Wind affecting the air volume flow through the tower or/and causing recirculation of air;
- (Temporary) pollution of sensors (e.g., dust on the ambient temperature sensor).

The applicability of the method was tested in two real cooling towers which were equipped with the respective sensors. In general, the model could be well calibrated also with their data, in more than 90% of the cases the deviation between modelled and measured cooling capacity was below 10%. For two out of three datasets the model-based evaluation over extended periods of time also showed reasonable results with variations like the calibration period. Yet, for one dataset a prolonged period with a significant and rather constant deviation and subsequent "return to normal" was observed. An explanation therefore could not be found in the data. This indicates that further testing of this method is necessary. An extension of the measurement technology and model for redundancy (e.g., measuring and calculating the evaporation loss) may also be helpful in order to assure that the measurements are correct. Finally, a coupling of the performance monitoring with the building control system may provide further valuable information on its operation mode.

If the described next steps are successful, the automated fouling detection has the potential to yield financial savings to cooling tower operators: it will allow cooling towers to be cleaned only when it is actually necessary because the performance is affected but before the impact of the fouling leads to a significant increase in operating costs or even undesired downtimes. As a positive side-effect, it would reduce the damage to the environment due to use of chemicals for premature cleaning or increased energy consumption due to fouling-related performance decrease.

Concluding, we found that it is challenging to provide reliable data with a minimum number of sensors. With increasing experience on this approach and by taking further influencing factors into consideration, this approach still seems promising for extended application in practice.

**Author Contributions:** Conceptualization, data curation and draft preparation: B.N.; methodology: B.N., M.M., A.S., K.C.; writing—review and editing: L.S. and K.C.; project administration: K.C. and B.N.; funding acquisition: K.C. and L.S. All authors have read and agreed to the published version of the manuscript.

**Funding:** This research was funded by the German Federal Ministry of Education and Research, grant number FKZ 01LY1618A/B.

**Institutional Review Board Statement:** Not applicable.

**Informed Consent Statement:** Not applicable.

**Data Availability Statement:** The data presented in this study are available on request from the corresponding author.

**Conflicts of Interest:** The authors declare no conflict of interest. The funders had no role in the design of the study; in the collection, analyses, or interpretation of data; in the writing of the manuscript, or in the decision to publish the results.

## Abbreviations

| Symbol | Unit | Description |
|---|---|---|
| $t_{w,in}$ | °C | Process fluid inlet temperature |
| $\dot{m}_w$ | kg/s | Process fluid mass flow rate |
| $\dot{m}_a$ | kg/s | Air mass flow rate |
| $R_{ext/int}$ | K/W | Air/process side heat resistance |
| $cp_{sat}$ | kJ/kg/K | Specific heat capacity of saturated air |
| $h_{a,in/out}$ | kJ/kg | Specific enthalpy of inlet/outlet air |

| $T_{wb,in/out}$ | K | Wet bulb temperature of inlet/outlet air |
| $UA_{ext/int}$ | W/K | Air/process side heat transfer coefficient |
| $\gamma$ | - | Constant parameter for external heat transfer |
| $\delta$ | W/m/K | Constant parameter for internal heat transfer |
| $\mu_w$ | kg/m/s | Dynamic viscosity of water |

**Appendix A**

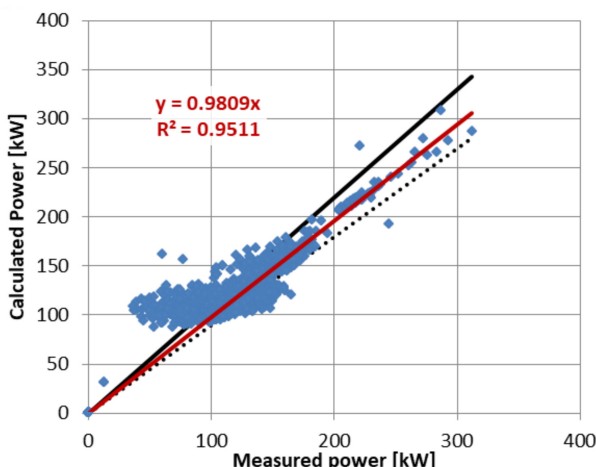

**Figure A1.** Result of model calibration with dataset CT1-Cal1, measured vs. calculated thermal power of cooling tower (41.2% outside +/−10% boundary); black solid/dotted lines show +/−10% boundaries.

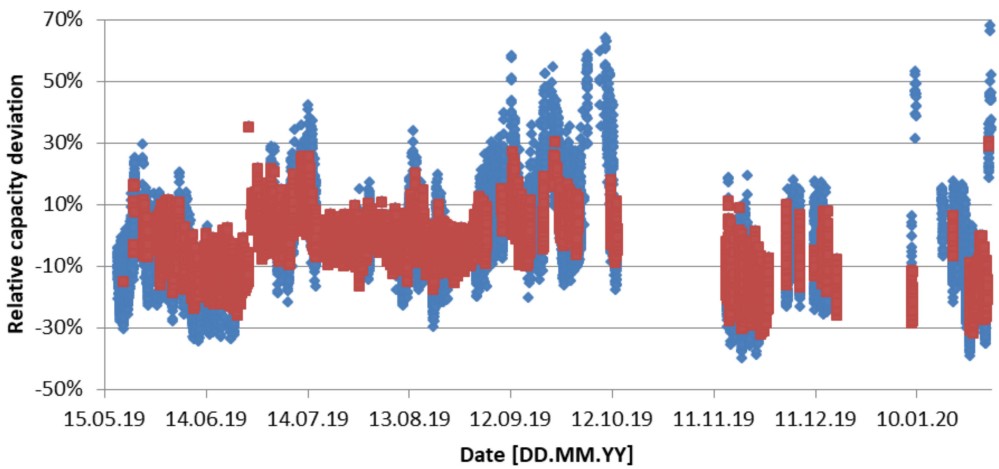

**Figure A2.** Trend of relative decrease of cooling tower power (calculated minus measures divided by measured) of dataset CT1-Eval2 calibrated with dataset CT2-Cal2; blue dots: all data, red dots: only 100% fan speed.

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
