# Peer review of "Model-Based Evaluation of Air-Side Fouling in Closed-Circuit Cooling Towers"

_energies, doi:10.3390/en14030695_

Round 1
Reviewer 1 Report
Nice work, pls give some more future perspective and applications for that approach…
general: passive voice would be more appropriate (18-21, 38-41, 348-351,...)
first time you use abbreviations pls announce it fully: 33 / GDP, 77 / CFU,
110: no bullet point?
235: cold system --> consider word change
fig 3 & 6 x Axis does not match description [DD.MM.YY] e.g. 1.3. --> 01.03. ...
Author Response
Dear Reviewer,
Thank you for your concise review – please find our answers to your comments (bullet points) below.
Best regards,
Björn Nienborg
- Nice work, pls give some more future perspective and applications for that approach…
Thank you! We extended the Conclusions accordingly.
- general: passive voice would be more appropriate (18-21, 38-41, 348-351,...)
We exchanged the 5 occurences of “we” with passive voice.
- first time you use abbreviations pls announce it fully: 33 / GDP, 77 / CFU,
Thanks for the hint, this should be self-evident – we exchanged the abbreviations for the full terms.
- 110: no bullet point?
Again, thanks for the hint, we corrected this mistake.
- 235: cold system --> consider word change
We changed the term to “cooling system”.
- fig 3 & 6 x Axis does not match description [DD.MM.YY] e.g. 1.3. --> 01.03. ...
Corrected the numbering of the axis in both graphs.
Reviewer 2 Report
This paper presents a model analysis method based on a closed cooling tower against fouling problems. Some comments are listed as followings:
- In the Abstract, it’s proposed to further simplify the background of the scaling problem while highlighting the innovativeness of this paper. The abstract now mainly presents the work of this paper, missing a summary of the main conclusions as well as innovative ones.
- In the Introduction, it is introduced that scaling of cooling towers will result in performance degradation, but the number of citations is little. It is suggested that more references should be made on cooling tower testing and scaling problem. It is also necessary to highlight the differences between the work of this paper and that of other researchers.
- In the Methodology section
- 1 needs some modification to check the symbol and position of the measuring point. In addition, the path of supplementary air, the position of the air measuring point and the measuring parameters are required. It is unclear whether air and water flow are counter-flow or parallel-flow.
- Only the temperature and moisture content of air are measured, but the air flow rate is not measured. As a result, the checking of heat balance between energy on the air side and energy on the water side is lacking.
- In Section 2.4, the calculation model of cooling tower is introduced, mainly based on literature [13-17]. However, these are the work of other researchers, so the innovative work in this paper cannot be seen. And the introduction of the model is simple. The topic of this paper is “Model-based evaluation of air-side fouling in closed-circuit cooling towers”, which requires more additional model details.
- Before the result analysis, the evaluation parameters in this paper and the calculation method of evaluation parameters need to be clarified. In the results analysis below, the reader may be confused as to “What these results mean? And how are these results calculated?”
- In the Results section
- 3 shows the change of fouling thickness of each pipe during measurement, which is intuitive. However, test conditions are not a single variable, but a combination of variables. It is suggested that different test periods be marked in Fig. 3 so that readers can understand the changes of test conditions.
- What does the horizontal and vertical coordinates in Fig. 5 represent and how they are calculated need to be supplemented.
- How the relative capacity deviation in Fig. 6 is calculated? And what is the physical meaning? Additional explanations are needed to explain the physical significance of the linear regression line, what does the slope of the regression line represent? The x of the regression line is the date?
- In Section 3.2.1, the relative capacity deviation of Fig. 11 is too large to reach 160%. What is the reason for the deviation? Does the large deviation mean that the applicability of this model is limited?
- In the Conclusions and Discussion sections. The conclusions are not concise and prominent enough to list the conclusions of the test and the model separately. Lines 329 to 334 explain the possible causes of divergence of test data, indicating that the applicability of this model is limited. It is recommended that these disturbed data be excluded for analysis.
Author Response
Dear Reviewer,
We thank you for your comprehensive review. Please find the answers to your comments (bullet points) below.
Best regards,
Björn Nienborg
- In the Abstract, it’s proposed to further simplify the background of the scaling problem while highlighting the innovativeness of this paper. The abstract now mainly presents the work of this paper, missing a summary of the main conclusions as well as innovative ones.
Thank you – we followed your advice.
- In the Introduction, it is introduced that scaling of cooling towers will result in performance degradation, but the number of citations is little. It is suggested that more references should be made on cooling tower testing and scaling problem. It is also necessary to highlight the differences between the work of this paper and that of other researchers.
We revised the introduction, included more references on fouling of cooling towers and pointed out the knowledge gap justifying this research paper.
In the Methodology section
- Figure 1 needs some modification to check the symbol and position of the measuring point. In addition, the path of supplementary air, the position of the air measuring point and the measuring parameters are required. It is unclear whether air and water flow are counter-flow or parallel-flow.
We admit that the scheme is not very detailed; we clarified the specified comments in the caption and the text.
- Only the temperature and moisture content of air are measured, but the air flow rate is not measured. As a result, the checking of heat balance between energy on the air side and energy on the water side is lacking.
Correct. As our objective is/was to develop a system that allows the performance monitoring (and subsequent fouling detection) with minimal interference of the regular system operation, we decided not to measure the air flow. In our experience the exact measurement of an air flow – allowing for an energy balance - is challenging and typically has a significant impact on the system itself due to the pressure drop.
- In Section 2.4, the calculation model of cooling tower is introduced, mainly based on literature [13-17]. However, these are the work of other researchers, so the innovative work in this paper cannot be seen. And the introduction of the model is simple. The topic of this paper is “Model-based evaluation of air-side fouling in closed-circuit cooling towers”, which requires more additional model details.
The novelty of this paper is that: a) we generate reliable measurement data of a cooling tower which is subject to fouling/scaling on a laboratory scale, b) we employ the presented model to detect the performance degradation due to fouling, c) we test the applicability of this approach to real installations based on the field measurements
Additionally, we refined the introduction and hope the scope and novelty of the content is clearer now.
- Before the result analysis, the evaluation parameters in this paper and the calculation method of evaluation parameters need to be clarified. In the results analysis below, the reader may be confused as to “What these results mean? And how are these results calculated?”
In our humble opinion, plotting the measured vs. the calculated values is quite a common way of evaluating the performance of a component model. We slightly adapted the description of figure 5a, with the intention to reduce the risk of confusion for the reader.
- In the Results section figure 3 shows the change of fouling thickness of each pipe during measurement, which is intuitive. However, test conditions are not a single variable, but a combination of variables. It is suggested that different test periods be marked in Fig. 3 so that readers can understand the changes of test conditions.
We measured the thickness 1-2 times per week while the operating conditions (controllable temperatures and flow rates, subsequently also cooling power) were varied various times a day; at this low time-resolution it would be rather overwhelming if not impossible to include this information in the graph.
- What does the horizontal and vertical coordinates in Fig. 5 represent and how they are calculated need to be supplemented.
We slightly extended the explanation of the graph in order to clarify this issue. As energies is an engineering journal, we chose not to include equations we presume are fundamental for possible readers (such as Q_dot = c_p * m_dot * (t_in – t_out)
- How the relative capacity deviation in Fig. 6 is calculated? And what is the physical meaning? Additional explanations are needed to explain the physical significance of the linear regression line, what does the slope of the regression line represent? The x of the regression line is the date?
The calculation and its interpretation are described in the text (starting lines 199). Correct, the date is the x of the regression line, therefore the slope is the relative capacity decrease per day.
- In Section 3.2.1, the relative capacity deviation of Fig. 11 is too large to reach 160%. What is the reason for the deviation? Does the large deviation mean that the applicability of this model is limited?
As we explain in the text, the calibration of the model is not optimal for this dataset because the water flow rates are significantly lower than during the operation period used for calibration. For this reason we included a second graph with a specific calibration for this dataset in the appendix A (Figure A2); here the relative capacity deviation is much lower, although the peaks of ~60% are still high; as these data points refer to the fan operating at low speed (blue dots), the measurements have a higher uncertainty and there is a higher risk of air recirculation etc. Additionally, as we mention in the text, there is an apparent offset in the results during a couple of summer weeks which we cannot explain with our measurements. This highlights the need for further verification of the approach, as we conclude.
- In the Conclusions and Discussion sections. The conclusions are not concise and prominent enough to list the conclusions of the test and the model separately. Lines 329 to 334 explain the possible causes of divergence of test data, indicating that the applicability of this model is limited. It is recommended that these disturbed data be excluded for analysis.
We would like to keep the structure of this section in order to be able to distinguish between the laboratory tests – which showed good results of the approach – and the “real life” tests – during which one out of three datasets put the approach into question and highlights the need for further verification. For this reason we decided to include also the “questionable” dataset and hope this is acceptable for you.
Reviewer 3 Report
Review: energies-1050778
Title: Model-based evaluation of air-side fouling in closed-2 circuit cooling towers
Authors: Björn Nienborg, Marc Mathieu, Alexander Schwärzler, Katharina Conzelmann, Lena Schnabel
General comments:
The paper is well written, complete, and exhaustive. The topic is interesting and the specific example can be of interest for applications to cooling towers.
The introduction is complete and detailed.
The methodology section is complete. The 1D model proposed is simple and could be effective if supported by valid experimental data. The only missing information is a detailed description of the Rint and Rext formulas. These formulas are necessary to understand the model implementation and to interpret the model behavior in the Results section.
The Results section is complete containing model tuning/validation tests and on-field applications. The results are interesting and well presented in several images.
The Discussion and Conclusions are sound and report the work findings in detail.
Some minor changes to be done have been highlighted and are reported in the pdf file attached together with some suggestions.
In particular:
- Thermal resistances in equations (1) and (2) should be reported explicitly.
- The sentence on page 8 lines 218-219 assumes that there is a linear dependence between scale layer thickness and performance decrease. Is that correct? Thermal resistance for a cylindrical layer depends on the natural log of the radiuses (external/internal) ratio. Was this accounted for? For the reviewer, it was not possible to verify the latter point from the information reported in the manuscript.
- Reference to Chapters should be changed to refer to Sections and the cross-links should be checked.

Author Response
Dear Reviewer,
we thank you for your comprehensive review. Please find the answers to your comments (bullet points) below.
Best regards,
Björn Nienborg
- Some minor changes to be done have been highlighted and are reported in the pdf file attached together with some suggestions.
Thank you for the detailed review – we incorporated your comments.
- Thermal resistances in equations (1) and (2) should be reported explicitly.
Done, we extended equations 2 and 3.
- The sentence on page 8 lines 218-219 assumes that there is a linear dependence between scale layer thickness and performance decrease. Is that correct? Thermal resistance for a cylindrical layer depends on the natural log of the radiuses (external/internal) ratio. Was this accounted for? For the reviewer, it was not possible to verify the latter point from the information reported in the manuscript.
Thank you – this is correct, of course. We added a comment on this fact on page 7 line 188, where we first do this simplified calculation (or better estimation).
- Reference to Chapters should be changed to refer to Sections and the cross-links should be checked.
Done (e.g. line 196).
Reviewer 4 Report
Review of the paper
Title: Model-based evaluation of air-side fouling in closed-circuit cooling
towers
By: Björn Nienborg, Marc Mathieu, Alexander Schwärzler, Katharina Conzelmann, Lena Schnabel
Manuscript number: energies-1020778
Submitted to: Energies
This paper presents a study on performance evaluation of air-side fouling in closed-circuit cooling towers. The topic is interesting. The scope of the journal covers this research matter. On its actual form, the paper looks more like a technical report rather than a scientific research paper. The whole paper should be re-written to include a comprehensive literature review, discussion on the used methodology, compare it with other methods, validation, and explanation and discussion on results. I have listed the main concerns. The following will guide the authors.
1. The authors have to improve the introduction; it does not provide sufficient background.
2. Please clarify the novelty of the submitted manuscript. Explain the value of this research.
3. The authors should include the specific objectives in the introduction.
4. In Section 2.4, provide the assumptions for Eqs. 1-3. Define the variables in Eqs. 1-3. Provide also their derivations.
5. In Section 2, provide a summary table with the input data and physical properties.
6. In Section 2.4 provide some details on the implementation of the cooling tower calculation as mentioned in Lines 128-131. Explain the L-BFGS-B algorithm and the scipy package.
7. In Sections 3 and 4, the analysis and discussion of results are poor. Authors should propose tentative explanation behind the described behaviour of their results.
8. Provide a table of nomenclature and acronyms.
9. Run the spelling tool for grammar/typing mistakes.
10. Improve the presentation of the figures.
Author Response
Dear Reviewer,
Thank you for your honest and critical comments! Please find our answers below.
Best regards,
Björn Nienborg
- The authors have to improve the introduction; it does not provide sufficient background.
We followed your advice and extended the introduction (section 1.) with further background from literature.
- Please clarify the novelty of the submitted manuscript. Explain the value of this research. / The authors should include the specific objectives in the introduction.
We conclude the revised introduction with the specific objectives of our work and its novelty.
- In Section 2.4, provide the assumptions for Eqs. 1-3. Define the variables in Eqs. 1-3. Provide also their derivations.
As the model is described in detail in the cited literature our intention was to merely summarize its essentials. Following your recommendation we decided to include some more information, still with the intention to keep this part short.
- In Section 2, provide a summary table with the input data and physical properties.
In our humble opinion, table 2 provides the relevant information.
- In Section 2.4 provide some details on the implementation of the cooling tower calculation as mentioned in Lines 128-131. Explain the L-BFGS-B algorithm and the scipy package.
As there is extensive information on the scipy package and the selected algorithm in the cited source and they are not specifically linked to the topic of the paper, we would like to refer the reader interested specifically in this topic to the reference.
- In Sections 3 and 4, the analysis and discussion of results are poor. Authors should propose tentative explanation behind the described behaviour of their results.
In our humble opinion, we do provide a concise and comprehensive interpretation of the results. If you have specific suggestions which aspects we should include we will be happy to consider them.
- Provide a table of nomenclature and acronyms.
Starting in line 422, there is a table of nomenclature; as the energies template does not include a space for it in the beginning of the paper, as one may find it familiar from other journals, we decided to include it in the appendix.
- Run the spelling tool for grammar/typing mistakes.
Done (American English).
- Improve the presentation of the figures.
We adapted especially figures 3 and 6.
Reviewer 5 Report
The following points must be addressed before publication:
- The abstract must to be improved. The Comparative-quantitative results should be mentioned in the Abstract.
- Please explain in detail what makes your study different from the available literature.
- It would be better for the authors to highlight the state of the art of the study in the last paragraph of the introduction. The objectives of the study is not clear. The author should state the objectives of the current study in the last paragraph in the introduction section. The last paragraph of the Introduction section should include:
- Precise aim and objectives of the paper.
- Limitations in the available literature. (research gap of related to present investigation)
- It would be better for the authors to arrange the summary of each section follows the sequence.
- More Comparative-quantitative results should be mentioned in the Conclusion.
- It is highly recommended to improve the English quality of the manuscript. There are a lot of grammatical errors.
Author Response
Dear Reviewer,
we thank you for your comprehensive review. Please find the answers to your comments (bullet points) below.
Best regards,
Björn Nienborg
- The abstract must to be improved. The Comparative-quantitative results should be mentioned in the Abstract.
Done, thank you.
- Please explain in detail what makes your study different from the available literature.
We revised the introduction and added a paragraph on the novelty of our research paper in the end of the introduction.
- It would be better for the authors to highlight the state of the art of the study in the last paragraph of the introduction. The objectives of the study is not clear. The author should state the objectives of the current study in the last paragraph in the introduction section. The last paragraph of the Introduction section should include:
- Precise aim and objectives of the paper.
- Limitations in the available literature. (research gap of related to present investigation)
We revised the introduction and followed your recommendations.
- More Comparative-quantitative results should be mentioned in the Conclusion.
In our view, the key results are summarized in the conclusions, yet we adapted the section in parts. If you are still missing information, please let us know specifically, we will be happy to consider incorporating it!
- It is highly recommended to improve the English quality of the manuscript. There are a lot of grammatical errors.
After completing the revision we once more used the built-in spell check of MS Word and had a native reader screen the document.
Round 2
Reviewer 4 Report
I have reviewed the revised paper. The literature review conducted does not fully cover the aspects of the subject area. The authors have addressed my comments/suggestions very succinctly.